# Using the Integrated Kano–RIPA Model to Explore Teaching Quality of Physical Education Programs in Taiwan

**DOI:** 10.3390/ijerph17113954

**Published:** 2020-06-03

**Authors:** Gordon Chih-Ming Ku, I-Wei Shang

**Affiliations:** 1Department of Social Sport, Lingnan Normal University, No.29, Cunjin Rd., Chikan Dist., Zhanjiang 524048, Guangdong, China; taiwanku520@gmail.com; 2Department of Physical Education and Kinesiology, National Dong Hwa University, No. 1, Sec. 2., Da-Hsueh Rd., Shou-Feng, Hualien 974, Taiwan

**Keywords:** student satisfaction, physical education programs, curricula design, teaching quality, revised importance–performance analysis

## Abstract

The purpose of this study was to apply the Kano model and revised importance–performance analysis (RIPA) to explore the teaching quality of physical education programs at Taiwanese universities. Random sampling was used to select universities from the north, south, central, and east areas of Taiwan. The questionnaire developed by the researcher included 20 items within four subscales. A total of 970 students participated in this study. A combination of the Kano model and RIPA was used for analysis. The results indicate that the item “physical education teachers have cordial attitudes toward instructions” fell into the “keep up the good work” designation of RIPA and the fourth quadrant of the Kano model. A patient and cordial attitude towards students can, therefore, be considered an important quality factor for physical education programs. Teacher–student relationships should, therefore, form a priority for physical education teachers looking to increase student satisfaction and optimize their programs.

## 1. Introduction

The education system can be viewed as a production system. Applying the concept of enterprise quality management to educational institutions, we can identify teaching quality as the primary factor influencing the production of graduating students whose habits and skillsets meet the needs of the economy and society [1,2]. A comprehensive review of educational research provides evidence that teaching quality maintains a direct causal relationship with the learning outcomes of students, including learning satisfaction [3], learning motivation [4], and academic performance [5]. Several studies on the teaching quality of higher-education institutions have found that teaching quality contributes to reducing behavioral problems [6,7]. Teaching quality and students’ learning satisfaction remain essential issues in higher-education research.

Physical education programs at all levels are responsible for helping students learn the value of physical activity for health, recreation, and social purposes. A core goal of the physical education curriculum is to actively explore personal health and fitness levels, as well as aid in the development of a life-long habit of regular exercise [8]. Performance indicators in studies on higher education have focused on research outputs; however, these studies usually do not take into account the teaching function of universities and colleges. To keep pace with the changing needs of students within a competitive environment, higher-education institutions must begin to focus on the current status of and strategies to improve teaching quality. At higher-education institutions, physical education courses gradually become elective. Several researchers have shown that as this happens, students’ selective motives, course values, and numbers of classes decline [9]. To improve enrolment rates in physical education courses, it is necessary to understand student needs and learning satisfaction. Research findings have shown that teaching quality is a significant factor influencing students’ levels of satisfaction with physical education programs [10,11,12].

Importance–performance analysis (IPA) is an effective evaluation tool for the design of improvement strategies for product–service systems [13,14]. IPA employs a two-subscale grid based on customer-perceived importance of service attributes and attributes performance. IPA is widely used as a research tool in fields such as service quality [15], tourism management [16,17], and park environmental management [18]. A handful of studies have applied this tool to research on higher-education institutions [19,20,21], but it has rarely been applied to research on physical education within these institutions. In this study, IPA is used to examine the priority ranking of teaching quality attributes, in order to enable the design of appropriate improvement strategies. 

However, the results of the traditional IPA have been doubted because of its two defeats. First, the dimensions of performance and importance are correlated. Second, performance indicators and customers’ satisfaction are not linear and nonsymmetrical [22]. These two defects may cause inaccuracy and misjudging the real situation of perceived product or service quality. Deng [23] suggested a revised IPA (RIPA) approach to solve the statistical problems of traditional IPA. The RIPA is conducted by several steps: (1) all performance indicators are transformed into natural logarithmic value; (2) every natural logarithmic value of performance indicator are calculated with overall satisfaction into partial correlation analysis, while the rest of performance indicators are fixed; (3) each partial correlation coefficient of performance indicator is calculated into absolute coefficient, which is the implicitly derived importance; (4) the performance indicators and the implicitly derived importance are used to draw the RIPA grid. The researcher believes that RIPA can provide more precise results to reflect the product or service performance. Accordingly, the present study adopts RIPA instead of traditional IPA to evaluate the teaching quality of physical education [23]. 

While RIPA identifies significant factors onto which customers attach importance and provides an effective framework for evaluation of performance, it offers little data on customer satisfaction levels. Recently, research has been performed on the asymmetric and nonlinear relationship between quality attributes and customer satisfaction using Kano’s two-subscales quality model. This study, therefore, also incorporated the Kano model of customer satisfaction [24]. The Kano model extends traditional service quality thinking; this model is a useful tool to understand customer needs by identifying and classifying the quality attributes [25]. The model classifies five types of quality requirements: attractive requirements (sufficient quality attributes lead to customer satisfaction, but insufficient quality attributes do not lead to customer dissatisfaction); one-dimension requirements (sufficient quality attributes lead to customer satisfaction, and insufficient quality attributes lead to customer dissatisfaction); must-be requirements (sufficient quality attributes do not lead to customer satisfaction, but insufficient quality attributes lead to customer dissatisfaction.); indifferent requirements (sufficient quality attributes do not lead to customer satisfaction, and insufficient quality attributes do not lead to customer dissatisfaction); reverse requirements (sufficient quality attributes lead to customer dissatisfaction, and insufficient quality attributes lead to customer satisfaction). Then, it calculates the “increase the average value of customer satisfaction coefficient” and “eliminate the average value of customer satisfaction coefficient”, the two mean crossings to distinguish between “height increase satisfaction, low eliminate dissatisfaction” (first quadrant), “low increase satisfaction, high eliminate dissatisfaction” (second quadrant), “low increase satisfaction, low eliminate dissatisfaction” (third quadrant) and “height increase satisfaction, height elimination of dissatisfaction” (fourth quadrant) [24]. Thus, the fourth quadrant of the Kano model is the most important indicator for improved efficiency. The indicators of teaching quality are also located in the “concentrate here” of the RIPA perception map and the fourth quadrant of the Kano model, which can be regarded as a key indicator in teaching quality.

In general, physical education programs adopt curriculum design, curriculum content, class management, and instructional guidance according to teaching quality assessment standards [26,27,28]. Research into teaching quality within physical education is often based on a conceptual model of service quality (SERVQUAL) by Parasuraman, Zeithaml and Berry (PZB), which includes the following attributes: reliability, reactivity, tangible, guaranteed and affinity [29,30]. However, there are some studies to escape the concept of PZB service quality to compose the quality of physical education teaching. Liu, Pan, and Chou [31] proposed combining the teaching quality model. They interviewed Taiwanese university students and found that the following four aspects were important indicators of teaching quality: “classroom management”, “teaching strategy”, “learning assessment”, and “course content”. Although there exists much research into the teaching quality of physical education programs in Taiwan [29,30,31,32], only one study [32] has examined this topic through the combined perspectives of IPA and the Kano model. These two approaches assess different aspects of quality; therefore, the use of the double mechanism provides a more comprehensive evaluation of customer requirements and current quality levels [33,34], thereby enabling the design of more effective improvement strategies. The purpose of this study was to use the Kano model to explore the student-perceived classification of attributes of teaching quality in physical education programs of Taiwanese universities. This study further analyzed teaching quality under the framework of RIPA. Finally, combining the two models allowed for the identification of the key indicators of teaching quality in university-level physical education programs.

## 2. Materials and Methods

### 2.1. Participants

In this study, Taiwan was divided into four regions: north, central, south, and east. The universities were randomly sampled from each region. Samples included all grades, from freshmen to seniors. All participants were informed that the study followed the ethical guidelines of National Science Council in Taiwan, so their responses would be confidential and used for research purposes only. In total, there were 539 female students (55.6%), and 431 male (44.4%). The northern sample comprised 468 (48.2%) students, 199 (20.5%)students made up the central sample; 167 (17.2%) students were selected from the south, and 136 (14%) students were selected from the east. 

### 2.2. Measures

This study directly adopted a scale of “University Students’ Perceived Satisfaction of Teaching Quality of Physical Education”, which was developed by Liu, Pan, and Chou [31]. The authors identify the key indicators of teaching quality of physical education from ungraduated students’ perspectives in Taiwan. The mixed-method was used to construct the scale. First, the study explored the potential indicators of teaching quality by interviewing three teachers and nine students, and utilized content analysis to find the items and dimensions of the scale. Afterward, a quantitative method was used to verify the structure of the scale. Seven hundred and twenty-four eligible respondents were recruited, and the collected data was analyzed by exploratory factor analysis (EFA), confirmatory factor analysis (CFA), and Cronbach’s *α*. The results indicated that the scale of teaching quality of physical education is comprised of 20 items in the following four subscales: classroom management (4 items), teaching strategy (5 items), learning assessment (5 items), and course content (6 items). In the EFA, these four subscales were with eigenvalue points of 2.73–3.59, and the cumulative variance of scale achieved 62.7%. The results of the CFA indicated a good fit (CFI = 0.98; SRMR = 0.03; RMSEA = 0.07) and the construct reliability ranged from 0.88 to 0.90. Furthermore, the average variance extracted ranged from 0.57 to 0.68. Cronbach’s α of the dimensions were respectively 0.95, 0.90, 0.93, 0.89 for classroom management, teaching strategy, learning assessment, and course content, and the total Cronbach’s alpha was 0.97.

### 2.3. Assessment of Teaching Quality Based on the Kano Model

The questionnaire of this study was developed based on the Kano model, in which each attribute was represented by two questions with responses drawn from the following five-point Likert-type scale: (1) I like it that way; (2) it must be that way; (3) I am indifferent; (4) I can live with it that way; (5) I dislike it that way. Responses were used to create the same 5 × 5 cross-match table as was used in the work of Matzler and Hinterhuber [24]. Answers to both the positive and negative subscales are cross-referenced. For example, if a participant responds, “I like it that way” to the positive subscale and “I can live with it that way” to the negative subscale, the service attribute is classified as “Attractive”. 

### 2.4. Assessment of Teaching Quality Based on Revised Importance–Performance Analysis 

The RIPA scale measures performance and overall satisfaction [35]. Performance refers to the indicators that can be considered as satisfied to teaching quality and are measured using the following five-point Likert scale: “1 = not at all satisfied”, “2 = not satisfied”, “3 = neutral”, “4 = satisfied”, “5 = very satisfied”. Overall satisfaction refers to the level of satisfaction experienced by students with regard to teaching quality, and is measured using the following five-point Likert scale: “1 = very dissatisfied”, “2 = dissatisfied”, “3 = neutral”, “4 = satisfied”, “5 = very satisfied”. For each item, the scores were averaged and then normalized for plotting on a RIPA perceptual grid.

Studying both RIPA and the Kano model simultaneously allowed this research to explore a deeper understanding of the key indicators involved in university students’ perceptions of the teaching quality of physical education. 

### 2.5. Statistical Analysis

The data collected by this study were analyzed using SPSS18.0 version software. Descriptive statistical analysis was applied to understand the distribution of demographic data. RIPA and the Kano model were used simultaneously to identify the critical indicators involved in the teaching quality of physical education in universities in Taiwan. Of the RIPA results, the mean scores of all indicators of teaching quality transformed into logarithmic value. Each indicator of natural logarithmic value was calculated with overall satisfaction into partial correlation analysis. The partial correlation coefficient of the indicators was revealed, which was regarded as the implicitly derived importance. The mean score and the implicitly-derived importance of the indicators were used to draw the RIPA grid. The RIPA grid is divided into four quadrants: “keep up the good work” (high performance–high importance), “concentrate here”(high importance–low performance), “low priority”(low performance–low importance), and “possible overkill”(low importance–high performance) (Figure 1). The indicators in the quadrant “keep up the good work” mean that the teaching quality is of high satisfaction and importance, and the advantage should be retained. The quadrant “concentrate here” represents that the teaching quality indicators are important but of low satisfaction. The teaching quality should be immediately improved. The quadrant “low priority” means that the teaching quality indicators are unimportant and unsatisfied. The indicators can be ignored temporarily. The final quadrant “possible overkill” is low importance but high satisfaction of teaching quality indicators. Teachers should shift the resources to areas that require improvement. 

The Kano model’s mean scores of increasing satisfaction and reducing dissatisfaction of all indicators were utilized to map the Kano model grid, which is consisted of three dimensions: basic factors (quadrant IV), excitement factors (quadrant II), and performance factors (quadrant I and III) (Figure 2). The basic factors are that the teaching quality indicators are dissatisfied because the teaching quality does not address students’ needs. However, the students would be satisfied when the indicators address their needs because it is the “basic requirement”. Therefore, addressing basic needs is necessary. The excitement factors represent that the indicators of teaching quality can cause students’ satisfaction, but it would not cause students’ dissatisfaction if the teaching quality does not address their needs. The indicators in excitement factors can be regarded as the extra bonus in the class. Finally, the performance factors are that the students feel satisfied if the expectation of teaching quality is fulfilled or exceeded, vice versa. Accordingly, the indicators located in the quadrant of “keep up with good work” and “concentrate here”, as well as those that fell into the quadrant “IV” in the Kano model are regarded as critical indicators of teaching quality. 

## 3. Results

### 3.1. RIPA Plots 

RIPA plot results are shown in as shown in Figure 3. “Physical education teachers have cordial attitudes toward instructions” and “The planning of the physical education curriculum is systematic” fell into the “keep up the good work” designation. These indicators of teaching quality must continue to maintain for promoting teaching quality in the physical education program; “The teaching material of the physical education curriculum meets students’ needs” fell into the “concentrate here” category. The indicator represents key areas that need to be improved with top priority for university students. “Physical education teachers create a good learning environment”, “Physical education teachers build confidence regarding students’ learning”, “Physical education teachers provide feedback for students’ questions”, “Physical education teachers set up an objective evaluation criteria” and “The content of physical education could be integrated with curriculum objectives” were designated “possible overkill”. These indicators do not affect teaching quality in physical education for university students. “Being a team member of a physical education class”, “Physical education teachers have innovative teaching strategies in a class”, “Physical education teachers motivate students’ interests”, “Physical education teachers guide the ability to think independently”, “Physical education teachers design group competition in a class”, “Physical education teachers use multiple methods to motivate students’ learning”, “Physical education teachers provide remedial teaching for students with poor learning outcomes”, “Physical education teachers measure students’ performance in an appropriate way”, “Physical education teachers assess the progress of students’ learning”, “The physical education curriculum guides sports values”, and “The physical education curriculum guides students to achieve a good learning attitude” were labeled “low priority”. These indicators were neither important nor poor in physical education program. According to the above results, most items of the teaching strategies aspects are in low priority. 

### 3.2. The Kano Model for Requirements Classification of Teaching Quality in Physical Education

As shown in Table 1, attributes important to teaching quality were categorized into one-dimension and indifferent requirements. None of the indicators fell into the must-be, attractive, and reverse requirements. The attributes relevant to each quality are listed in the following: 

One-dimension requirements: The following one-dimension requirements were identified in each of the four subscales: (1) classroom management—attitude toward instructions, ability to create a good learning environment, ability to build students’ self-confidence; (2) teaching strategy—none; (3) learning assessment—willingness to provide feedback, appropriate evaluation strategies, provision of objective evaluation criteria; (4) course content—integration of course content with curriculum objectives, provision of appropriate guidance for learning attitudes through curriculum design, appropriate design of activities, ability of curriculum design to meet students’ needs. The provision of these indicators ensures student satisfaction with teaching quality.

Indifferent requirements: The following attributes were identified as indifferent requirements in each of the four subscales: (1) classroom management—encouragement of teamwork; (2) teaching strategy—innovative teaching strategies, ability to motivate students, encouragement of independent thinking, encouragement of a sense of competitiveness, diverse approach to teaching strategies; (3) learning assessment—none; (4) course content—systematic curriculum design, encouragement to value physical activity through curriculum design. The results suggest that the above attributes were not important to students. 

### 3.3. Customer Satisfaction Coefficients for the Kano Model Grid

Following the classification of the attributes of teaching quality, the customer satisfaction (CS) coefficient, which is indicative of how strongly a service attribute may affect customer satisfaction, or in the case, of its nonfulfillment and customer dissatisfaction [36,37]. The formula to calculate the average impact of a service attribute on satisfaction is defined as CS(1) = (A+O)/(A+O+M+I). The CS (1) values range from 0 to 1. The formula to calculate the average impact of that on dissatisfaction is defined as CS (2) = (O+M)/(A+O+M+I). When the average value of an increased customer satisfaction coefficient is closer to 1, the customers perceive higher satisfaction. When the average value of a decreased customer dissatisfaction coefficient is closer to –1, customers perceive lower dissatisfaction. The CS (1) and CS (2) values can be plotted against the X- and Y-axes, respectively, of a Kano model grid [37].

For CS (1), the range of values fell between 0.56 and 0.39. The top five indicators were as follows: the attitude of instruction (0.56), willingness to provide feedback (0.49), ability to create a good learning environment (0.47), design of objective evaluation criteria (0.47), and encouragement of good learning attitude through curriculum design (0.46). For CS (2), the range of values fell between 0.48 and 0.66. The top five indicators were as follows: the attitude of instruction (−0.66), ability to create a good learning environment (–0.60), design of objective evaluation criteria (−0.60), willingness to provide feedback (−0.60), and ability to build students’ self-confidence (−0.59). The CS (1) and CS (2) values are respectively plotted against the X- and Y-axes of a Kano grid. The overall means of the CS (1) and CS (2) values are treated as the central line in the grid, as shown in Figure 4. Factors of teaching quality are allocated to quadrant IV when they hold significant influence. Factors included in this quadrant are “cordial attitude towards students”, “ability to create a good learning environment”, “ability to build student self-confidence”, “willingness to provide feedback”, and “ability to design objective evaluation criteria”. Factors in quadrant IV have the greatest impact on students’ satisfaction with physical education programs at Taiwanese universities.

### 3.4. Applying an Integrated Kano—IPA Model to Identify Attributes Key to Teaching Quality of Physical Education

The following aspects were located in the fourth quadrant of the Kano model: cordial attitude towards students, ability to create an excellent learning environment, ability to build students’ self-confidence, willingness to provide feedback, and ability to design an objective evaluation criteria. A cordial attitude towards students and systematic curriculum design are located in “keep up the good work” of IPA. An attribute can be designated as a critical attribute for improvement when it lands in these sections of the two models (Table 2). Thus, a cordial attitude towards students was highlighted as the highest priority for physical education teachers looking to improve their teaching quality.

## 4. Discussion

RIPA analysis identified 12 attributes of teaching quality that are of low priority in terms of improvement; that is, the performance of these attributes was weak, but the importance of these attributes was low. The majority of these fell into the subscales of teaching strategy. Our results indicate that teaching strategy is no longer a vital indicator to improve students’ satisfaction. This finding is inconsistent with those of previous studies, as most physical education teachers emphasize technical skills and teaching strategies [38]. Physical education teachers attempt to improve learning outcomes by focusing on teaching skills and imparting knowledge rather than fulfilling students’ needs. When physical education teachers do not design their curricula to meet students’ needs, their motivation to participate in physical education is adversely affected. A shift to student-centered learning would help in this regard [39], as only courses that are up-to-date, interesting, and pertinent to students’ needs will attract enrolment. Our results further indicate that physical education teachers at universities should allocate more resources to improving their attitudes toward instructions and the systematic nature of curriculum design.

The results produced by the Kano model confirm that students are indifferent to teaching strategies; that is, they do not affect students’ satisfaction with physical education programs. The following five attributes were found to be influential concerning student satisfaction, increasing students’ satisfaction and reducing students’ dissatisfaction: classroom management was highlighted as an area in need of attention; teacher attitudes, learning environment, and teacher–student relationships were identified as crucial to student satisfaction. This is consistent with previous studies [30,40]. When applying the Kano model in the study of teaching quality of physical education, Lee et al. [30] found that a right attitude, happy atmosphere, and sufficient communication increase student satisfaction. 

Factors located in both the fourth quadrant of the Kano grid and the “keep up the good work” of RIPA were designated as key to the improvement of physical education programs. The results emphasized teachers’ attitudes toward instruction, which not only improves satisfaction but also promotes learning motivation. The findings of previous studies support this. Lovorn et al. [40] indicated that humor positively influences interaction patterns and classroom social structure. Creating a good student–teacher relationship shapes the informal communication structure in a classroom. Positive student–teacher relationships increase motivation [41], enhance academic achievement, and reduce disciplinary problems [42]. The majority of research into the role of teachers’ interpersonal behavior within physical education has found that it can promote students’ motivation to engage in physical activity [43,44,45]. In Taiwan, several studies have found that a friendly attitude and positive environment positively influence students’ learning motivation [30]. This study’s integrated approach allows for more precise identification of the indicators key to improving teaching quality in physical education, enabling a more effective allocation of resources [33,34].

In the studies of Díaz-Méndez and Gummesson [46] and Lindsay, Breen, and Jenkins [47], it was mentioned that middle school students and university students rank teaching quality attributes differently, and a student’s age influences their perception of teaching quality. This study confirms the importance of classroom management (precisely teacher attitudes, learning environments, and teacher–student relationships) for university students. Chen, Ku, and Ku [32] found that middle schools were most concerned with facilities and equipment. The size of the class also affected middle-school students’ satisfaction with physical education teaching. In Taiwan, physical education courses in universities are not compulsory, and teachers design their curriculums autonomously. Physical education teachers in universities must have professional knowledge and skills, but further require personal characteristics, affinity, and ability to create a positive learning environment to affect students’ learning outcomes.

## 5. Conclusions

In the past, the core values of physical education were teacher-oriented, but recent years have seen a gradual shift towards a more student-oriented learning environment. A sensitive understanding of students’ needs is a practical approach to improving students’ motivation to enroll in physical education programs. Increased competition among higher-education institutions in Taiwan has increased the importance of student satisfaction. Using the Kano–RIPA integration model allows for the development of nuanced and targeted suggestions for physical education courses, which in turn enables the effective allocation of limited resources for a maximum effect on student satisfaction.

## Figures and Tables

**Figure 1 ijerph-17-03954-f001:**
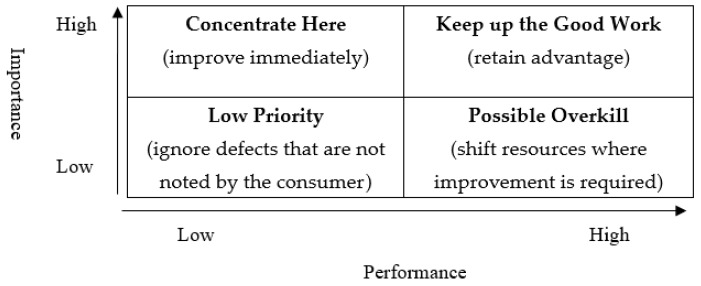
Revised importance–performance analysis (RIPA) Perceptual Map.

**Figure 2 ijerph-17-03954-f002:**
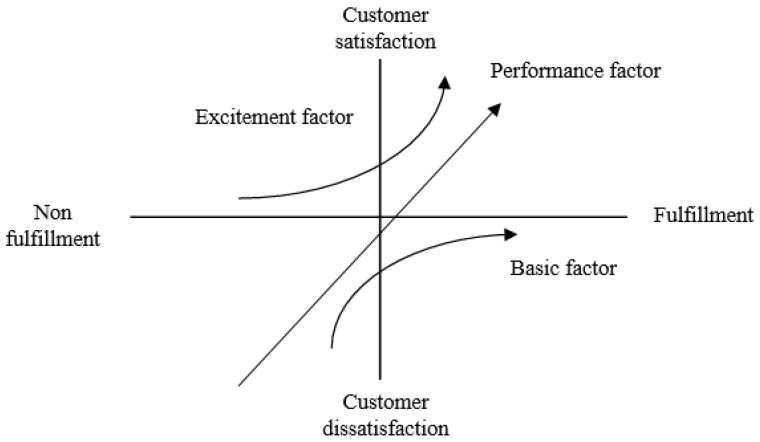
Kano Model.

**Figure 3 ijerph-17-03954-f003:**
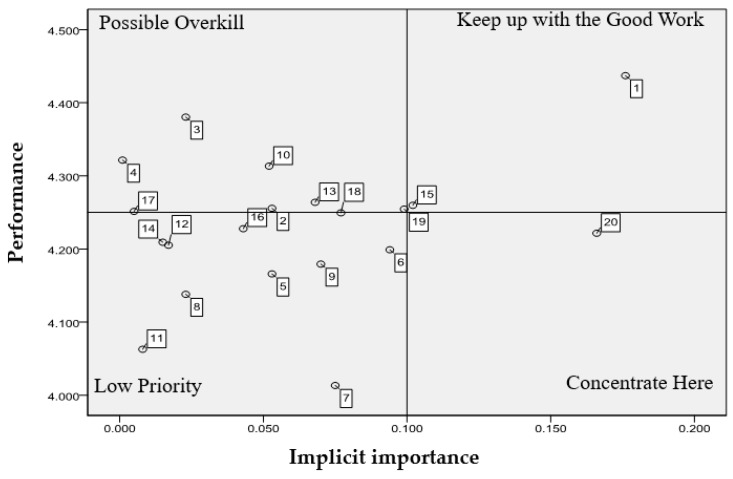
RIPA grid—teaching quality indicators of physical education.

**Figure 4 ijerph-17-03954-f004:**
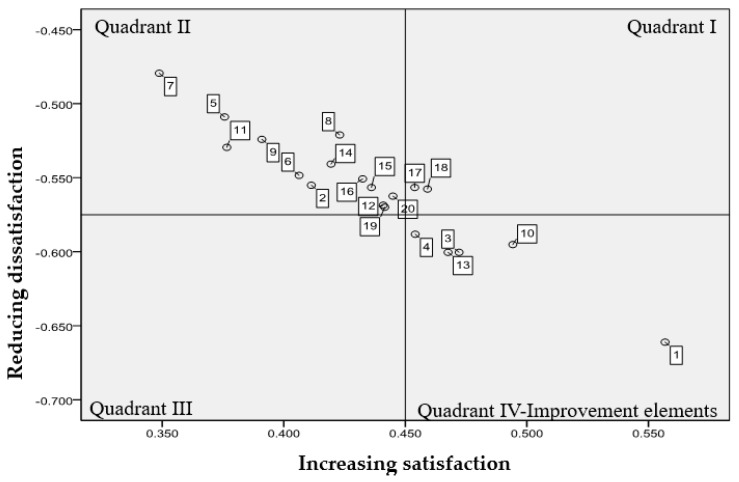
Kano model grid—teaching quality indicators of physical education.

**Table 1 ijerph-17-03954-t001:** Attribute classification of teaching quality of physical education on the Kano model.

Subscales	Items	Teaching Quality Requirements of the Kano Model Attribute Classification Weight %						Requirement Classification
		A	O	M	I	R	Q	
**Class management**	Physical education teachers have cordial attitudes towards instructions	6.39	48.10	16.60	26.80	0.21	1.86	O
	Being a team member of a physical education class	5.05	35.36	19.18	38.66	0.31	1.44	I
	Physical education teachers create a good learning environment	5.98	40.31	18.56	33.20	0.31	1.65	O
	Physical education teachers build confidence regarding students’ learning	6.30	38.08	19.40	33.95	0.62	1.65	O
Teaching strategy	Physical education teachers have innovative teaching strategies in a class	5.46	31.13	18.45	42.37	0.72	1.86	I
	Physical education teachers motivate students’ interest in sports	6.39	33.40	20.31	37.84	0.31	1.75	I
	Physical education teachers guide the ability to think independently	6.19	27.94	18.97	44.74	0.52	1.65	I
	Physical education teachers design group competition in a class	6.60	34.74	16.19	40.21	0.62	1.65	I
	Physical education teachers use multiple methods to motivate students’ learning	5.46	32.99	18.56	41.34	0.10	1.55	I
Learning assessment	Physical education teachers provide feedback for students’ questions	7.01	41.44	16.91	32.68	0.52	1.44	O
	Physical education teachers provide remedial teaching for students with poor learning outcomes	4.74	32.06	19.69	41.24	0.52	1.75	I
	Physical education teachers measure students’ performance in an appropriate way	5.46	37.63	17.94	36.70	0.52	1.75	O
	Physical education teachers set up an objective evaluation criteria	5.57	40.52	18.66	33.81	0.31	1.13	O
	Physical education teachers assess the progress of students’ learning	6.39	34.95	18.35	38.87	0.52	0.93	I
Course content	The planning of the physical education curriculum is systematic	4.54	38.35	16.39	39.07	0.62	1.03	I
	The physical education curriculum guide sports values	5.57	37.01	17.22	38.66	0.41	1.13	I
	The content of physical education could be integrated with curriculum objectives	5.77	38.87	15.88	37.84	0.41	1.24	O
	The physical education curriculum guide students to achieve a good learning attitude	6.80	38.35	16.49	36.70	0.41	1.24	O
	Physical education teachers plan appropriate physical fitness activities	5.88	37.73	18.56	36.60	0.21	1.03	O
	The teaching material of the physical education curriculum meets students’ needs	5.57	38.14	17.11	37.42	0.41	1.34	O

A = attractive quality; O = one-dimension quality; M = must-be quality; I = indifferent quality; R = reverse quality; Q = questionable.

**Table 2 ijerph-17-03954-t002:** Integrated RIPA and Kano model cross-analysis on teaching quality of physical education.

Items	Indicators of Teaching Quality of Physical Education	Kano Quadrant	RIPA
1	Physical education teachers have cordial attitudes toward instructions	4	Keep up the good work
2	Being a team member of a physical education class	2	Low priority
3	Physical education teachers create a good learning environment	4	Possible overkill
4	Physical education teachers build confidence regarding students’ learning	4	Possible overkill
5	Physical education teachers have innovative teaching strategies in a class	2	Low priority
6	Physical education teachers motivate students’ interest in sports	2	Low priority
7	Physical education teachers guide the ability to think independently	2	Low priority
8	Physical education teachers design group competition in a class	2	Low priority
9	Physical education teachers use multiple methods to motivate students’ learning	2	Low priority
10	Physical education teachers provide feedback for students’ questions	4	Possible overkill
11	Physical education teachers provide remedial teaching for students with poor learning outcomes	2	Low priority
12	Physical education teachers measure students’ performance in an appropriate way	2	Low priority
13	Physical education teachers set up an objective evaluation criteria	4	Possible overkill
14	Physical education teachers assess the progress of students’ learning	2	Low priority
15	The planning of the physical education curriculum is systematic	2	Keep up the good work
16	The physical education curriculum guides sports values	2	Low priority
17	The content of physical education could be integrated with curriculum objectives	1	Possible overkill
18	The physical education curriculum guide students to achieve a good learning attitude	1	Low priority
19	Physical education teachers plan appropriate physical fitness activities	3	Low priority
20	The teaching material of the physical education curriculum meets students’ needs	1	Concentrate here

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
