# Peer review of "Using the Integrated Kano–RIPA Model to Explore Teaching Quality of Physical Education Programs in Taiwan"

_ijerph, 2020, doi:10.3390/ijerph17113954_

Round 1
Reviewer 1 Report
Dear authors, first of all I would like to apologize for my English, it is not very correct, but I did not want to miss the opportunity to review this interesting paper. I think it would be very beneficial to physical education to increase student satisfaction and optimize their programs in college as well, because it could make a big difference in teacher-student relationships. Even though, the manuscripts present some flaws that must be considered. I recommend its publication after minor changes.
- In the abstract, why only present choices north and south? (The abstract show that random sampling was used to select universities from the north and south, but in L118, The northern sample 118 comprised 468 (48.2%) students, 199 (20.5%) students made up the central sample; 167 (17.2%) 119 students were selected from the South; and 136 (14%) students were selected from the East.)
- The results of the CFA indicated a good fit (SRMR =0.31). Is it wrongly planted? please could you check that they are correct.(Values for the SRMR range from zero to 1.0 with well-fitting models obtaining values less than .05 (Byrne, 1998; Diamantopoulos and Siguaw, 2000), however values as high as 0.08 are deemed acceptable (Hu and Bentler, 1999)) .
- It is mandatory to inform the approval of the ethics committee; the approval number of the ethical permission is as much as possible
Author Response
Dear Editor,
Thank you very much for forwarding to us the valuable comments from the reviewers. Their time and effort are very much appreciated. We have revised our manuscript based on their comments, and have summarized the changes in the following table:
Reviewer’s Comments and Changes Made |
Section & Page No. |
Reviewer 1 Dear authors, first of all I would like to apologize for my English, it is not very correct, but I did not want to miss the opportunity to review this interesting paper. I think it would be very beneficial to physical education to increase student satisfaction and optimize their programs in college as well, because it could make a big difference in teacher-student relationships. Even though, the manuscripts present some flaws that must be considered. I recommend its publication after minor changes.
Response and Changes Made Thank you for your suggestion to improve the abstract. We have added central and east area of Taiwan in the “Abstract” section (p.1) as follows: The purpose of this study was to apply the Kano model and Revised Importance-Performance Analysis (RIPA) to explore the teaching quality of physical education programs at Taiwanese universities. Random sampling was used to select universities from the north, south, central, and east areas of Taiwan. The questionnaire developed by the researcher included 20 items within four subscales. A total of 970 students participated in this study. A combination of the Kano model and RIPA was used for analysis. The results indicate that the item physical education teachers have a cordial attitude toward instruction fell into the keep up the good work designation of the RIPA and Quadrant IV of the Kano model. A patient and cordial attitude towards students can, therefore, be considered an important quality factor for physical education programs. Teacher-student relationships should, therefore, form a priority for physical education teachers looking to increase student satisfaction and optimize their programs.
|
“Abstract” (p.1) |
Response and Changes Made Thank you for pointing out this mistake. We have checked the original paper; the value of SRMR is 0.03. The value of SRMR has been revised in the “Materials and Methods” section (p.3) as follows: The original instrument designed by Liu, Pan, and Chou [(Liu, Pan, & Chou, 2015)] to measure university students’ perceived satisfaction of teaching quality of physical education. The authors adopted qualitative and quantitative method to find the potential items and dimensions by interview three teachers and nines students, and to establish the validity and reliability of questionnaire by recruiting 724 respondents. Item analysis, exploratory factor analysis (EFA), confirmatory factor analysis (CFA), and Cronbach’s α were used to analyze the data collected. The results indicated that the scale of teaching quality of physical education is comprised 20 items in the following four subscales: classroom management (4 items), teaching strategy (5 items), learning assessment (5 items), and course content (6 items). In EFA, these four subscales were with eigenvalue points of 2.73-3.59, and the cumulative variance of scale achieved 62.7%. The results of the CFA indicated a good fit (CFI = 0.98; SRMR =0.03; RMSEA =0.07), the construct reliability ranged from 0.88 to 0.90. and the average variance extracted ranged from 0.57 to 0.68. Cronbach's α of the dimensions were respectively 0.95, 0.90, 0.93, 0.89 for classroom management, teaching strategy, learning assessment, and course content and the total Cronbach's alpha was 0.97.
|
“Materials and Methods” (p.3) |
Response and Changes Made Thank you for your suggestion. We did not apply for ethical approval from the ethics committee since we decided to use a quantitative method. All respondents are voluntarily and anonymously participated in the study. In the first part of the questionnaire, the informed consent was used to explain the purpose of this study, the respondents' right, and the declaration of no conflict of interest to the eligible respondents. If they agree with the content of informed consent, they would fill out the questionnaire, vice versa.
|
|

Reviewer 2 Report
Dear Authors:
The subject of study, ‘Assessing the quality of teaching of Physical Education Programmes’, seems interesting to me. The paper is well organized and it includes appropriate sections: introduction, methods, analysis, etc. The authors have made a good synthesis of the literature.
However, management concepts are mixed with education concepts, which is confusing. It is also difficult to understand if the Kano-RIPA models are not previously known. Why and how did they decide to adapt them specifically to the physical activity area?. On the other hand, although the study is aimed to students, authors talk about customers at different times. Could you clarify this aspect please?
In the methodology section I expected to find the instruments from Kano-Ripa. On the contrary, it is included an instrument designed by Liu, Pan, and Chou [31] to measure university students ’perceived satisfaction of teaching quality of physical education’. This reference is from 2015. In addition, this instrument is confusing and difficult to understand for me. It should be better explained. I think that this paper represent a good job, but unfortunately it is very unclear in some aspects for me. Thus, the research work that has been carried out is very difficult to understand just by reading the text.
The authors also noted: ‘Although there is much research into the teaching quality of physical education programs in Taiwan [29-31], few studies have examined this topic through the combined perspectives of IPA and Kano model’. Therefore, Did the authors really analyse the same as many studies using another model?
Regarding the statistical analysis, it can be found in the text: ‘Descriptive statistical analysis was applied to understand the distribution of demographic data, while RIPA and Kano model were used to jointly identify the key elements involved in the teaching quality’. Thus, has only a descriptive analysis been done?
According to the use of English, some mistakes have been detected in the manuscript. Review by an expert proofreader in the area is an interesting option to consider.
In summary, I find the outcomes and the research subject interesting. However, this paper is not sufficiently specific for this journal. Choosing an education or management magazine may be more appropriate.
My final recommendation is to revise the text in depth to improve general understanding, and to send the article to a specialized journal in topics as education or management.
Author Response
Dear Editor,
Thank you very much for forwarding to us the valuable comments from the reviewers. Their time and effort are very much appreciated. We have revised our manuscript based on their comments, and have summarized the changes in the following table:
Reviewer’s Comments and Changes Made |
Section & Page No. |
Reviewer 2 Dear Authors: The subject of study, ‘Assessing the quality of teaching of Physical Education Programmes’, seems interesting to me. The paper is well organized and it includes appropriate sections: introduction, methods, analysis, etc. The authors have made a good synthesis of the literature. However, management concepts are mixed with education concepts, which is confusing. It is also difficult to understand if the Kano-RIPA models are not previously known. Why and how did they decide to adapt them specifically to the physical activity area? On the other hand, although the study is aimed to students, authors talk about customers at different times. Could you clarify this aspect please?
Response and Changes Made Thank you for your comments. The management concepts applied to education fields have been proved to contribute to improving teachers’ teaching quality and students’ learning outcomes. IPA and Kano model are useful approaches in the business management for improving the service quality. These two approaches have been used to evaluate teaching quality to improve students’ learning satisfaction in the educational field. For example, Alberty and Mihalik (1989) used the IPA approach to evaluate teaching quality in adult education. Cho (2012) adapted the Kano model to explore the factors that influence college students’ learning satisfaction. On the other hand, RIPA is a better approach instead of a traditional IPA. To identify the key indicators of teaching quality, this study selected RIPA combined with the Kano model to analyze the data. Kuo, Chang, and Lai (2011) mentioned that “an integrated approach that combines the Kano model and IPA can help to solve the challenge of exploring the critical elements of education” (p. 12018). According to the foundation of these two approaches, the term “customer” was only used to explain the RIPA and Kano model concepts. We thought to explain these two approaches with the original concept that would be clear and easy to understand their application. Afterward, we further illustrated how these two approaches apply to teaching quality in physical education. For example, “First, the dimensions of performance and importance are correlated. Second, performance indicators and customers’ satisfaction are not linear and non-symmetrical [22]” (p.2). That explains how IPA works in the research. I hope we do answer your question clearly. If you have further questions or comments, please let us know.
|
“Abstract” (p.1) |
In the methodology section I expected to find the instruments from Kano-RIPA. On the contrary, it is included an instrument designed by Liu, Pan, and Chou [31] to measure university students ’perceived satisfaction of teaching quality of physical education’. This reference is from 2015. In addition, this instrument is confusing and difficult to understand for me. It should be better explained. I think that this paper represents a good job, but unfortunately it is very unclear in some aspects for me. Thus, the research work that has been carried out is very difficult to understand just by reading the text.
Response and Changes Made Thank you for your comments. We have further explained the instrument of this study. Moreover, the RIPA and Kano model have also been interpreted in the “Materials and Method” section (pp.3-4) as follows: This study directly adopted a scale of "University Students' Perceived Satisfaction of Teaching Quality of Physical Education, which was developed by Liu, Pan, and Chou [31]. The authors identify the key indicators of teaching quality of physical education from ungraduated students' perspectives in Taiwan. The mixed-method was used to construct the scale. First, the study explored the potential indicators of teaching quality by interview three teachers and nine students, and utilized content analysis to find the items and dimensions of the scale. Afterward, a quantitative method was used to verify the structure of the scale. Seven hundred and twenty-four eligible respondents were recruited, and the collected data was analyzed by exploratory factor analysis (EFA), confirmatory factor analysis (CFA), and Cronbach's α. The results indicated that the scale of teaching quality of physical education is comprised of 20 items in the following four subscales: classroom management (4 items), teaching strategy (5 items), learning assessment (5 items), and course content (6 items). In EFA, these four subscales were with eigenvalue points of 2.73-3.59, and the cumulative variance of scale achieved 62.7%. The results of the CFA indicated a good fit (CFI = 0.98; SRMR =0.03; RMSEA =0.07), the construct reliability ranged from 0.88 to 0.90. Furthermore, the average variance extracted ranged from 0.57 to 0.68. Cronbach's α of the dimensions were respectively 0.95, 0.90, 0.93, 0.89 for classroom management, teaching strategy, learning assessment, and course content, and the total Cronbach's alpha was 0.97. 2.3. Assessment of Teaching Quality based on Kano model The questionnaire of this study was developed based on the Kano model, in which each attribute was represented by two questions with responses drawn from the following five-point Likert-type scale: (1) I like it that way; (2) it must be that way; (3) I am indifferent; (4) I can live with it that way; (5) I dislike it that way. Responses were used to create the same 5 x 5 cross-match table as was used in the work of Matzler and Hinterhuber [24]. Answers to both the positive and negative subscales are cross-referenced. For example, if a participant responds “I like it that way” to the positive subscale and “I can live with it that way” to the negative subscale, the service attribute is classified as “Attractive”.
2.4. Assessment of Teaching Quality based on Revised Importance-Performance Analysis The RIPA scale measures performance and overall satisfaction [35]. Performance refers to the elements that can be considered as satisfied to teaching quality and are measured using the following five-point Likert scale: "1=not at all satisfied", "2=not satisfied", "3=neutral", "4= satisfied", "5=very satisfied". Overall satisfaction refers to the level of satisfaction experienced by students with regard to teaching quality, and is measured using the following five-point Likert scale: "1=very dissatisfied", "2=dissatisfied" "3=neutral", "4=satisfied", "5=very satisfied". For each item, the scores were averaged and then normalized for plotting on a RIPA perceptual grid. Studying both the RIPA and Kano model simultaneously allowed this research to explore a deeper understanding of the critical indicators involved in university students’ perceptions of the teaching quality of physical education. The indicators both located in the quadrant of "keep up with good work" and "concentrate here", and fell into the quadrant of "improvement elements" in Kano model are regarded as critical indicators of teaching quality.
|
“Materials and Methods” (pp.3-4) |
The authors also noted: ‘Although there is much research into the teaching quality of physical education programs in Taiwan [29-31], few studies have examined this topic through the combined perspectives of IPA and Kano model’. Therefore, Did the authors really analyse the same as many studies using another model?
Response and Changes Made Thank you for pointed out the unclear sentence. We have been rewritten the sentence in the “Introduction” section (p.3). The studies on teaching quality of physical education are a lot in Taiwan. However, only one combined two models to analyze the teaching quality of physical education from junior high school students’ perspectives. The study provided insight information and useful suggestions for the curriculum design of physical education for junior high school. Therefore, this study used the same approach to explore the teaching quality of physical education in universities.
In general, physical education programs adopt curriculum design, curriculum content, class management, and instructional guidance according to teaching quality assessment standards [26-28]. Research into teaching quality within physical education is often based on a conceptual model of service quality (SERVQUAL) by Parasuraman, Zeithaml, & Berry (PZB), which includes the following attributes: reliability, reactivity, tangible, guaranteed and affinity [29, 30]. However, there are some studies to escape the concept of PZB service quality to compose the quality of physical education teaching. Liu, Pan, and Chou [31] proposed combined the teaching quality model. The study was interviewed Taiwanese university students and found that the following four aspects were important indicators of teaching quality: "classroom management", "teaching strategy", "learning assessment", and “course content”. Although there exists much research into the teaching quality of physical education programs in Taiwan [29-32], only one study [32] has examined this topic through the combined perspectives of IPA and Kano model. These two approaches assess different aspects of quality; therefore, the use of the double mechanism provides a more comprehensive evaluation of customer requirements and current quality levels [33, 34], thereby enabling the design of more effective improvement strategies. The purpose of this study was to use Kano model to explore the student-perceived classification of attributes of teaching quality in physical education programs of Taiwanese universities. This study further analyzed teaching quality under the framework of RIPA. Finally, combining the two models allowed for the identification of the key elements of teaching quality in university-level physical education programs.
|
“Introduction” (p.3) |
Regarding the statistical analysis, it can be found in the text: ‘Descriptive statistical analysis was applied to understand the distribution of demographic data, while RIPA and Kano model were used to jointly identify the key elements involved in the teaching quality’. Thus, has only a descriptive analysis been done?
Response and Changes Made Thank you for pointed out the unclear sentence. We have modified the sentence in the “Materials and Method” section (pp.3-4) as follows 2.5. Statistical analysis The data collected by this study were analyzed using SPSS18.0 version software. Descriptive statistical analysis was applied to understand the distribution of demographic data. RIPA and Kano model were used to identify the critical indicators involved in the teaching quality of physical education in universities in Taiwan.
|
“Materials and Methods” (pp.3-4) |
According to the use of English, some mistakes have been detected in the manuscript. Review by an expert proofreader in the area is an interesting option to consider.
Response and Changes Made Thank you for pointed out the grammar problems. We have improved the grammar and readability in the manuscript.
|
|
In summary, I find the outcomes and the research subject interesting. However, this paper is not sufficiently specific for this journal. Choosing an education or management magazine may be more appropriate. My final recommendation is to revise the text in depth to improve general understanding, and to send the article to a specialized journal in topics as education or management.
Response and Changes Made Thank you for your suggestion. Actually, we submit the article to International Journal of Environment Research and Public Health because of the special issue: Measurement and Evaluation in Physical Education, Physical Activity and Sports. We think the article is suitable for the special issue in IJERPH.
|
|

Reviewer 3 Report
The article is well written and structured. Also, author/authors show a correct use of research methodology. Findings are interesting and whorthy to be furtherly developed.
Author Response
Dear Editor,
Thank you very much for forwarding to us the valuable comments from the reviewers. Their time and effort are very much appreciated. We have revised our manuscript based on their comments, and have summarized the changes in the following table:
Reviewer’s Comments and Changes Made |
Section & Page No. |
Reviewer 3 The article is well written and structured. Also, author/authors show a correct use of research methodology. Findings are interesting and worthy to be furtherly developed.
Response and Changes Made Thank you for your support. I hope this paper will be published in the International Journal of Environmental Research and Public Health soon. Thank you!
|
|

Round 2
Reviewer 2 Report
My main concern is in "Statistical analysis". I still have doubts if statistically you have only done a descriptive study. You indicate that "RIPA and Kano model were used to identify the critical indicators". I wonder are they statistical models?
Author Response
Dear Editor,
Thank you very much for forwarding to us the valuable comments from the reviewers. Their time and effort are very much appreciated. We have revised our manuscript based on their comments, and have summarized the changes in the following table:
Reviewer’s Comments and Changes Made |
Section & Page No. |
Reviewer 2 My main concern is in "Statistical analysis". I still have doubts if statistically you have only done a descriptive study. You indicate that "RIPA and Kano model were used to identify the critical indicators". I wonder are they statistical models?
Response and Changes Made Thank you for your comment. We have further explained the statistical analysis about RIPA and Kano model in the “Materials and Methods” (p.4) as follows: 2.5. Statistical analysis The data collected by this study were analyzed using SPSS18.0 version software. Descriptive statistical analysis was applied to understand the distribution of demographic data. RIPA and Kano model were used to identify the critical indicators involved in the teaching quality of physical education in universities in Taiwan. Of the RIPA, the mean scores of all indicators of teaching quality transformed into logarithmic value. Each indicator of natural logarithmic value was calculated with overall satisfaction into partial correlation analysis. The partial correlation coefficient of the indicators was revealed, which was regarded as the implicitly derived importance. The mean score and the implicitly derived importance of the indicators were used to draw the RIPA grid. The Kano model's mean scores of increasing satisfaction and reducing dissatisfaction of all indicators were utilized to map the Kano model grid. The indicators both located in the quadrant of "keep up with good work" and "concentrate here", and fell into the quadrant " IV" in the Kano model are regarded as critical indicators of teaching quality.
|
“Materials and Methods” (p.4)
|
